# Predictors of severe strongyloidiasis and mortality in hospitalized patients from Southern Thailand

Sorawit Chittrakarn ⓘ*, Nonthanat Tongsengkee, Nattapat Sangkakul, Siripen Kanchanasuwan

Division of Infectious Disease, Department of Internal Medicine, Faculty of Medicine, Prince of Songkla University, Songkhla, Thailand

* Sorawit.c@psu.ac.th

## Abstract

### Background

Strongyloidiasis is a neglected tropical infection that can progress to life-threatening disease in immunocompromised hosts. However, predictors of severe disease and mortality remain incompletely defined, particularly in endemic regions. We aimed to identify clinical predictors of severe strongyloidiasis and 30-day mortality among hospitalized patients in southern Thailand.

### Methods

We conducted a 16-year retrospective cohort study of hospitalized patients aged ≥13 years with laboratory-confirmed *Strongyloides stercoralis* infection at a tertiary-care hospital between 2009 and 2025. Severe strongyloidiasis was defined by larval detection in extraintestinal or respiratory specimens or compatible clinical manifestations involving multiple organ systems with infectious complications. Multivariable logistic regression was used to identify predictors of severe disease and mortality.

### Results

Among 735 patients, 47 (6.4%) had severe strongyloidiasis. Corticosteroid exposure within 60 days was independently associated with severe disease (adjusted odds ratio [aOR] 3.25, 95% CI 1.65–6.73), whereas eosinophilia (absolute eosinophil count ≥500 cells/μL) was associated with lower odds (aOR 0.41, 95% CI 0.19–0.82). Bacteremia was more frequent in severe cases (38.3% vs 4.1%, p < 0.001), predominantly due to gram-negative organisms. Severe strongyloidiasis was associated with higher 30-day mortality (38.3% vs 14.0%, p < 0.001). In multivariable analysis, increasing age (aOR 1.22 per 10-year increase), higher National Early Warning Score (aOR 1.41 per point), and bacteremia (aOR 3.38) were independently associated with

**Data availability statement:** All relevant data are within the manuscript and its Supporting information files.

**Funding:** The author(s) received no specific funding for this work.

**Competing interests:** The authors have declared that no competing interests exist.

mortality, whereas eosinophilia was associated with lower odds of death and severe strongyloidiasis itself was not.

## Conclusions

Corticosteroid exposure was the strongest predictor of severe strongyloidiasis, while eosinophilia was protective. Mortality was driven primarily by bacteremia and physiologic instability. Early recognition and treatment are essential to improve outcomes in endemic settings.

### Author summary

Strongyloidiasis is a parasitic infection common in tropical regions that can become life-threatening in immunocompromised individuals. However, clinical predictors of severe disease and death are not well defined, particularly in endemic settings. In this 16-year study of hospitalized patients in southern Thailand, recent corticosteroid exposure was strongly associated with severe strongyloidiasis. In contrast, the presence of eosinophilia was independently associated with a lower risk of both severe disease and 30-day mortality. Severe cases were frequently complicated by gram-negative bacteremia and had higher short-term mortality. However, mortality overall was primarily driven by older age, physiologic instability at presentation, and bacteremia rather than severe strongyloidiasis alone. These findings clarify key clinical factors associated with adverse outcomes and may assist clinicians in risk stratification of hospitalized patients in endemic areas.

## Introduction

Strongyloidiasis, caused by *Strongyloides stercoralis*, is a neglected tropical parasitic infection with a unique ability to autoinfect the human host, allowing the parasite to persist for decades without external re-exposure [1]. The infection is endemic in many tropical and subtropical regions and remains an important cause of morbidity and mortality in low- and middle-income countries [2]. Although most infected individuals remain asymptomatic or develop only mild gastrointestinal or dermatologic symptoms, the disease can progress to a severe, life-threatening form when host immunity is compromised [3].

Current knowledge regarding epidemiology, risk factors, and clinical outcomes of severe strongyloidiasis remains limited. Much of the available literature consists of case reports and small case series [4]. Such data are inherently vulnerable to publication bias and under-reporting, as dramatic or fatal presentations are more likely to be described, while milder or undiagnosed cases may remain unrecognized. This limitation is particularly evident in Southeast Asia, where systematically collected data on severe strongyloidiasis are scarce despite the region's endemicity.

The clinical diagnosis of strongyloidiasis is challenging. The disease lacks pathognomonic clinical features, especially in hospitalized patients who often have multiple comorbidities and concurrent infections. In severe cases, manifestations such as sepsis, respiratory failure, or gastrointestinal symptoms may be indistinguishable from other causes. Although hyperinfection and disseminated strongyloidiasis are traditionally distinguished, these entities frequently overlap in clinical practice, and definitive confirmation of dissemination often requires invasive diagnostic procedures that are rarely feasible. Therefore, pragmatic classification approaches may better reflect real-world settings [5].

Given these diagnostic and epidemiologic limitations, there remains a need for large, systematically collected cohort data to better define predictors of severe disease and short-term mortality in hospitalized patients. Such data are essential to improve early recognition and risk stratification, particularly in endemic settings where the disease burden may be underestimated.

To address these gaps, we conducted a 16-year retrospective cohort study of hospitalized patients with laboratory-confirmed strongyloidiasis at a tertiary-care center in southern Thailand. The objectives were to identify clinical predictors of severe strongyloidiasis and to determine factors independently associated with 30-day mortality.

## Methods

### Ethics statement

The study was conducted in accordance with the Declaration of Helsinki, and the protocol was approved by the Prince of Songkla university Ethics Committee of REC.68-528-14-1 on 18 December 2025 with a waiver of informed consent due to the retrospective nature of the study.

### Study design and setting

This retrospective cohort study was conducted at Songklanagarind Hospital, a 1,000-bed university-affiliated tertiary-care center in southern Thailand. The hospital serves as the primary referral center for the southern region of the country. All hospitalized patients with laboratory-confirmed *Strongyloides stercoralis* infection between January 1, 2009 and October 31, 2025 were identified through the hospital electronic database in collaboration with the Digital Innovation and Data Analytics (DIDA) unit.

### Study population

Eligible participants were hospitalized patients aged ≥13 years with laboratory-confirmed strongyloidiasis, defined by direct visualization of *Strongyloides stercoralis* larvae in any clinical specimen (e.g., stool, sputum, bronchoalveolar lavage, gastric aspirate, sterile body fluid, or tissue biopsy).

Only the first hospitalization episode with confirmed strongyloidiasis during the study period was included for each patient. Patients with incomplete medical records were excluded.

### Case definitions

Patients were classified as having uncomplicated or severe strongyloidiasis based on systematic chart review.
Severe strongyloidiasis was defined as either:

1. Direct visualization of *Strongyloides stercoralis* larvae in extraintestinal or respiratory specimens, or

2. Laboratory-confirmed strongyloidiasis with acute clinical manifestations involving at least two of the following domains:

1) Systemic symptoms (e.g., fever, chill, sepsis, septic shock)

2) Gastrointestinal symptoms (e.g., acute diarrhea, ileus, abdominal pain)

3) Respiratory symptoms (e.g., dyspnea, hypoxemia, new pulmonary infiltrates, respiratory failure)

4) Cutaneous manifestations (e.g., larva currens, petechiae, purpura)

5) Infectious complications temporally associated with diagnosis (e.g., bacteremia without an alternative source, CNS infection)

AND a clinical diagnosis of severe strongyloidiasis documented by an infectious diseases specialist.

Uncomplicated strongyloidiasis was defined as laboratory-confirmed infection without clinical or microbiologic evidence of severe disease.

Because clinical manifestations of hyperinfection and dissemination frequently overlap in routine practice, and definitive proof of dissemination often requires invasive diagnostic procedures that are not routinely performed, cases were pragmatically categorized as severe versus uncomplicated for clinically meaningful analysis.

### Data collection and outcomes

Demographic characteristics, comorbidities, immunosuppressive exposures, clinical manifestations, laboratory findings, radiographic results, treatment regimens, and outcomes were extracted from electronic medical records.

Immunosuppressive exposure was defined as receipt of corticosteroids, chemotherapy, or other immunosuppressive agents within 60 days prior to diagnosis. Physiologic severity at diagnosis was assessed using the National Early Warning Score (NEWS). Vasopressor use was recorded if administered on the day of diagnosis. Bacteremia was defined as a positive blood culture occurring from 2 days before to 7 days after the date of diagnosis. Laboratory parameters were recorded on the day of diagnosis; if unavailable, the closest values obtained within the preceding 48 hours were used.

The primary outcome was development of severe strongyloidiasis. Secondary outcomes included in-hospital mortality, 14-day mortality, 30-day mortality, length of hospital stay, ICU admission within 7 days of diagnosis, vasopressor use within 7 days, and mechanical ventilation within 7 days. Mortality status was confirmed through hospital records and cross-checked with the national death registry when available.

### Statistical analysis

Continuous variables were summarized as medians with interquartile ranges (IQRs) and compared using the Wilcoxon rank-sum test. Categorical variables were expressed as frequencies and percentages and compared using the chi-square test or Fisher's exact test, as appropriate. Multivariable logistic regression analysis was performed to identify factors independently associated with severe strongyloidiasis and factors associated with 30-day mortality. Variables considered clinically relevant or demonstrating $P < 0.10$ in univariable analysis were evaluated for inclusion, with final model selection guided by biologic plausibility and model stability. Adjusted odds ratios (aORs) with 95% confidence intervals (CIs) were reported. Model discrimination was assessed using the area under the receiver operating characteristic curve (AUC). All statistical analyses were performed using RStudio. Missing data were handled by complete-case analysis. The overall extent of missing data was minimal across all study variables, as clinical, laboratory, and outcome data were systematically extracted from electronic medical records; the proportion of missing values was low for key model variables, as laboratory parameters were recorded on the day of diagnosis or within the preceding 48 hours per protocol. A two-sided p value <0.05 was considered statistically significant.

## Results

### Baseline characteristics and risk factors

A total of 735 hospitalized patients with laboratory-confirmed strongyloidiasis were included in the study. Of these, 47 patients (6.4%) met criteria for severe strongyloidiasis, while 688 (93.6%) were classified as uncomplicated. The median age of the cohort was 63.0 years (IQR 51.5–71.7), and 81.9% were male.

Baseline characteristics are summarized in Table 1. Age and sex distribution did not differ significantly between groups. Most underlying comorbidities were comparable between patients with severe and uncomplicated disease. However, corticosteroid use within 60 days prior to diagnosis was significantly more common among patients with severe strongyloidiasis (72.3% vs 46.2%, p<0.001). Chronic kidney disease was less frequent in the severe group (0% vs 10.9%, p=0.032). Other immunosuppressive exposures, including chemotherapy and other immunosuppressive agents, were not significantly different between groups.

## Clinical and laboratory findings at diagnosis

Clinical and laboratory findings at diagnosis are presented in **Table 2**. Patients with severe disease demonstrated greater physiologic instability, with higher median temperature, heart rate, respiratory rate, and NEWS score, as well as lower systolic blood pressure compared with those with uncomplicated infection (all p<0.05). Vasopressor use at diagnosis was significantly more frequent in the severe group (14.9% vs 3.5%, p=0.001), whereas ICU admission at diagnosis did not differ significantly (17.0% vs 10.3%, p=0.233).

Total leukocyte and absolute neutrophil counts were similar between groups. In contrast, the absolute eosinophil count was significantly lower in patients with severe disease (median 142 vs 341 cells/μL, p=0.021), and eosinophilia (AEC ≥ 500 cells/μL) was less common among severe cases (21.3% vs 40.3%, p=0.015). Moreover, moderate to severe eosinophilia was observed exclusively in the uncomplicated group (p=0.033). Bacteremia occurred substantially more often in severe cases (38.3% vs 4.1%, p<0.001), predominantly due to gram-negative organisms.

Among patients with uncomplicated strongyloidiasis, the most common indication for stool examination was evaluation of nosocomial diarrhea, accounting for 295 of 688 patients (42.9%). Screening prior to initiation of immunosuppressive therapy was the second most frequent indication (110 patients, 16.0%), followed by evaluation of eosinophilia (109 patients, 15.8%). Stool examination was also performed in patients with community-acquired diarrhea (66 patients, 9.6%). Less common indications included unclear or undocumented reasons (45 patients, 6.5%), incidental or other clinical

**Table 1. Baseline characteristics and risk factors of uncomplicated vs severe strongyloidiasis.**

| Variable | Uncomplicated (n=688) | Severe (n=47) | P-value |
|---|---|---|---|
| Age, years, median (IQR) | 62.9 (51.4–71.8) | 63.1 (52.6–70.8) | 0.742 |
| Male sex, n (%) | 565 (82.1) | 37 (78.7) | 0.697 |
| **Underlying diseases, n (%)** | | | |
| Diabetes mellitus, n (%) | 109 (15.8) | 9 (19.1) | 0.695 |
| Chronic kidney disease, n (%) | 75 (10.9) | 0 (0.0) | 0.032 |
| COPD, n (%) | 57 (8.3) | 5 (10.6) | 0.771 |
| Chronic heart disease, n (%) | 124 (18.0) | 6 (12.8) | 0.474 |
| Cirrhosis, n (%) | 48 (7.0) | 3 (6.4) | 1.000 |
| Solid malignancy, n (%) | 220 (32.0) | 15 (31.9) | 1.000 |
| Hematologic malignancy, n (%) | 117 (17.0) | 12 (25.5) | 0.198 |
| Solid organ transplant, n (%) | 4 (0.6) | 0 (0.0) | 1.000 |
| HIV infection, n (%) | 10 (1.5) | 2 (4.3) | 0.383 |
| Autoimmune disease, n (%) | 63 (9.2) | 6 (12.8) | 0.574 |
| Alcoholism, n (%) | 41 (6.0) | 1 (2.1) | 0.441 |
| **Immunosuppressive agents (≤60 days), n (%)** | | | |
| Corticosteroid use | 318 (46.2) | 34 (72.3) | <0.001 |
| Chemotherapy | 142 (20.6) | 11 (23.4) | 0.790 |
| Other immunosuppressive agents | 19 (2.8) | 3 (6.4) | 0.333 |

**Table 2. Clinical features, physiologic severity markers and laboratory findings of uncomplicated versus severe strongyloidiasis.**

| | Uncomplicated (n = 688) | Severe (n = 47) | P-value |
|---|---|---|---|
| **Hospital course at diagnosis** | | | |
| ICU admission, n (%) | 71 (10.3) | 8 (17.0) | 0.233 |
| Vasopressor use, n (%) | 24 (3.5) | 7 (14.9) | 0.001 |
| Days of hospitalization before diagnosis, median (IQR) | 6 (2–12) | 5 (2–9) | 0.287 |
| **Physiologic severity markers at day of diagnosis** | | | |
| Maximum temperature, °C, median (IQR) | 37.4 (37.0–38.2) | 38.1 (37.5–39.3) | <0.001 |
| Maximum pulse rate, beats/min, median (IQR) | 93 (80–107) | 117 (96–124) | <0.001 |
| Maximum respiratory rate, breaths/min, median (IQR) | 24 (20–26) | 26 (24–30) | <0.001 |
| Minimum systolic BP, mmHg, median (IQR) | 110 (100–120) | 100 (95–116) | 0.002 |
| Minimum diastolic BP, mmHg, median (IQR) | 63 (60–70) | 62 (51–65) | 0.033 |
| NEWS score, median (IQR) | 3 (2–5) | 6 (5–8) | <0.001 |
| **Laboratory findings at diagnosis** | | | |
| WBC (×10³/µL), median (IQR) | 9.7 (6.9–13.9) | 8.7 (5.2–14.4) | 0.278 |
| ANC (cells/µL), median (IQR) | 6,449 (4,175–10,580) | 7,005 (3,529–12,115) | 0.936 |
| AEC (cells/µL), median (IQR) | 341 (138–796) | 142 (50–418) | 0.021 |
| Any eosinophilia (AEC ≥ 500 cells/µL), n (%) | 277 (40.3) | 10 (21.3) | 0.015 |
| Eosinophilia severity, n (%) | | | |
| Mild (500–1,499 cells/µL) | 207 (30.1) | 10 (21.3) | 0.033 |
| Moderate (1,500–5,000 cells/µL) | 63 (9.2) | 0 (0.0) | — |
| Severe (>5,000 cells/µL) | 7 (1.0) | 0 (0.0) | — |
| **Diagnosis** | | | |
| Stool examination, n (%) | 688 (100) | 45 (95.7) | — |
| Non-stool gastrointestinal specimen*, n (%) | – | 5 (10.6) | |
| Respiratory specimen†, n (%) | – | 21 (44.7) | |
| Other sterile body fluid specimen‡, n (%) | – | 2 (4.3) | |
| **Bacteremia** | | | |
| Any bacteremia, n (%) | 28 (4.1) | 18 (38.3) | <0.001 |
| Gram-negative bacteremia, n (%) | 21 (3.1) | 17 (36.2) | <0.001 |
| Polymicrobial bacteremia, n (%) | 6 (0.9) | 4 (8.5) | <0.001 |

\* Includes gastric aspirate, small bowel aspirate, and gastrointestinal biopsy.

†Includes sputum, bronchoalveolar lavage (BAL), and lung biopsy.

‡Includes pleural fluid and abdominal collection.

Abbreviations: ICU; intensive care unit, ANC; absolute neutrophil count, AEC; absolute eosinophil count.

evaluations (41 patients, 6.0%), suspected parasitic infection other than *Strongyloides stercoralis* (12 patients, 1.7%), and cancer-related screening or diagnostic workup (10 patients, 1.5%). However, the presence of diarrhea as an indication for testing does not establish a direct causal relationship between *Strongyloides* infection and diarrhea, as nosocomial diarrhea in hospitalized patients is often multifactorial.

Among patients with severe strongyloidiasis, stool examination was positive in 95.7% of cases. Respiratory specimens were positive in 44.7%, non-stool gastrointestinal specimens in 10.6%, and other sterile-site specimens in 4.3%. Detailed clinical characteristics of patients with severe strongyloidiasis, including organ involvement, radiographic findings, bacteremia characteristics, and specimen positivity, are provided in **S1-S4 Tables**.

Treatment patterns and early clinical course are shown in **Table 3**. All patients with severe strongyloidiasis received antiparasitic therapy. Combination therapy with oral albendazole plus ivermectin was significantly more common in severe

**Table 3. Antiparasitic treatment, early clinical course, and outcomes of uncomplicated versus severe strongyloidiasis.**

| Variable | Uncomplicated (n = 688) | Severe (n = 47) | P-value |
|---|---|---|---|
| **Antiparasitic treatment** | | | |
| Any antiparasitic treatment, n (%) | 658 (95.6) | 47 (100) | 0.138 |
| Oral Albendazole only, n (%) | 506 (73.5) | 12 (25.5) | <0.001 |
| Oral Ivermectin only, n (%) | 54 (7.8) | 6 (12.8) | 0.245 |
| Oral Albendazole + ivermectin, n (%) | 98 (14.2) | 29 (61.7) | <0.001 |
| **Time to antiparasitic treatment ≤48 h, n (%)** | 597 (86.8) | 47 (100) | 0.003 |
| **Duration of antiparasitic therapy, days, median (IQR)** | | | |
| Albendazole | 4.5 (2–7) | 8 (5–16) | <0.001 |
| Ivermectin | 2 (2–4) | 12 (7–17) | <0.001 |
| **Early clinical course after diagnosis** | | | |
| ICU admission after diagnosis, n (%) | 48 (7.0) | 16 (34.0) | <0.001 |
| Vasopressor use after diagnosis, n (%) | 19 (2.8) | 11 (23.4) | <0.001 |
| **Clinical outcomes** | | | |
| In-hospital mortality, n (%) | 28 (4.1) | 5 (10.6) | 0.053 |
| 14-day mortality, n (%) | 56 (8.1) | 14 (29.8) | <0.001 |
| 30-day mortality, n (%) | 96 (14.0) | 18 (38.3) | <0.001 |
| Length of hospital stay, days, median (IQR) | 16 (9–30) | 23 (12.5–38.5) | 0.024 |

disease (61.7% vs 14.2%, p<0.001), whereas albendazole monotherapy predominated in uncomplicated cases (73.5% vs 25.5%, p<0.001). At our institution, only oral antiparasitic agents were available during the study period. Thiabendazole was not available, and intravenous or subcutaneous ivermectin was not used, even in patients with severe disease.

Time to antiparasitic treatment within 48 hours was more frequent among severe cases (100% vs 86.8%, p=0.003). The duration of ivermectin therapy was markedly longer in severe disease (median 12 vs 2 days, p<0.001), as was albendazole therapy (8 vs 4.5 days, p<0.001). Within 7 days after diagnosis, ICU admission (34.0% vs 7.0%, p<0.001) and vasopressor use (23.4% vs 2.8%, p<0.001) were significantly more common among patients with severe disease.

Severe strongyloidiasis was associated with worse short-term outcomes. Fourteen-day mortality was significantly higher in the severe group (29.8% vs 8.1%, p<0.001), as was 30-day mortality (38.3% vs 14.0%, p<0.001). In-hospital mortality showed a higher proportion in the severe group (10.6% vs 4.1%), although this did not reach statistical significance (p=0.053). The median length of hospital stay was longer among patients with severe disease (23 vs 16 days, p=0.024).

In multivariable logistic regression analysis (**Table 4**), corticosteroid exposure within 60 days (aOR 3.25, 95% CI 1.65–6.73, p<0.001) was independently associated with severe strongyloidiasis, whereas the presence of eosinophilia (AEC ≥ 500 cells/μL) was independently associated with lower odds of severe disease (aOR 0.41, 95% CI 0.19–0.82, p=0.016). Age, sex, malignancy, and chemotherapy were not independently associated with severe disease. The model demonstrated acceptable discrimination, with an area under the receiver operating characteristic curve (AUC) of 0.69.

In the multivariable model evaluating 30-day mortality (**Table 5**), increasing age (aOR 1.22 per 10-year increase, 95% CI 1.04–1.43, p=0.015), higher NEWS score at diagnosis (aOR 1.41 per point increase, 95% CI 1.28–1.56, p<0.001), and bacteremia (aOR 3.38, 95% CI 1.63–7.02, p=0.001) were independently associated with mortality. In contrast, eosinophilia was independently associated with lower odds of 30-day mortality (aOR 0.46, 95% CI 0.27-0.79, p=0.005). Severe strongyloidiasis itself was not independently associated with 30-day mortality after adjustment (aOR 1.21, 95% CI 0.56–2.62, p=0.63). The 30-day mortality model showed good discrimination, with an AUC of 0.74.

**Table 4. Multivariable predictors of severe strongyloidiasis.**

| Predictor | Adjusted OR | 95% CI | P-value |
|---|---|---|---|
| Age (per 10-year increase) | 1.00 | 0.98-1.03 | 0.666 |
| Male sex | 0.97 | 0.47–2.16 | 0.934 |
| Corticosteroid use ≤ 60 days | 3.25 | 1.65–6.73 | **<0.001** |
| Chemotherapy | 0.61 | 0.26–1.33 | 0.224 |
| Hematologic malignancy | 1.76 | 0.79–3.73 | 0.15 |
| Solid malignancy | 1.02 | 0.49–2.06 | 0.954 |
| Presence of eosinophilia (AEC ≥ 500) | 0.41 | 0.19–0.82 | **0.016** |

**Table 5. Multivariable predictors of 30-day mortality.**

| Predictor | Adjusted OR | 95% CI | P-value |
|---|---|---|---|
| Severe strongyloidiasis (vs non-severe) | 1.21 | 0.56–2.62 | 0.63 |
| Age (per 10-year increase) | 1.22 | 1.04–1.43 | **0.015** |
| Male sex | 0.95 | 0.54–1.68 | 0.86 |
| Corticosteroid use ≤ 60 days | 0.97 | 0.61–1.55 | 0.90 |
| Any eosinophilia (AEC ≥ 500 cells/µL) | 0.46 | 0.27–0.79 | **0.005** |
| NEWS score at diagnosis (per 1-point increase) | 1.41 | 1.28–1.56 | **<0.001** |
| Bacteremia (yes) | 3.38 | 1.63–7.02 | **0.001** |

## Discussion

In this 16-year hospital-based cohort from southern Thailand, we identified clinical predictors of severe strongyloidiasis and 30-day mortality among hospitalized patients with parasitologically confirmed infection. Severe disease occurred in 6.4% of cases and was independently associated with recent corticosteroid exposure, whereas the presence of eosinophilia was associated with lower odds of severe disease. For 30-day mortality, increasing age, greater physiologic instability at diagnosis, and bacteremia were independently associated with death, while severe strongyloidiasis itself was not independently predictive after adjustment.

The prevalence of severe strongyloidiasis in our cohort (6.4%) was higher than previously reported estimates from population-based and healthcare database studies, which suggest that approximately 0.9–2% of infected individuals develop severe disease [6,7]. Several factors may explain this difference. First, our study included exclusively hospitalized patients with parasitologically confirmed infection, representing a clinically higher-risk population than community-based cohorts or studies relying primarily on serologic diagnosis. Second, Thailand has a relatively high background prevalence of strongyloidiasis, and hospitalized patients frequently receive corticosteroids or other immunosuppressive therapies that increase the risk of progression to severe disease [8–10]. Third, uncomplicated infections may be substantially underdiagnosed in our setting because direct stool examination, the primary diagnostic modality used, has limited sensitivity [11,12]. This limitation may preferentially detect patients with higher larval burden or more clinically apparent disease, thereby inflating the observed proportion of severe cases among confirmed diagnoses [13].

The distribution of established risk factors in our cohort was broadly consistent with global data, including male predominance and frequent underlying malignancy, chemotherapy exposure, and immunosuppressive therapy [14]. However, the proportion of patients with recent corticosteroid exposure was notably high, highlighting its central role as a precipitating factor in endemic settings. The lower prevalence of CKD among severe cases (0% vs 10.9%) is likely explained by the relatively small number of CKD patients in this cohort (n = 75) and the low absolute number of severe cases overall (n = 47), which limits the probability of observing CKD in the severe group by chance alone. CKD is not an established independent

risk factor for severe strongyloidiasis, and this finding should not be interpreted as a protective effect. HTLV-1, a recognized risk factor for severe strongyloidiasis in some endemic regions, was not identified in our cohort [15], consistent with the very low HTLV-1 prevalence in Thailand [16].

Consistent with prior literature, recent corticosteroid exposure was strongly associated with severe disease [4,14]. Corticosteroids impair host immune control by suppressing Th2-mediated responses and reducing eosinophil activity, which facilitates parasite survival and dissemination [17]. In addition, corticosteroids may enhance parasite replication by promoting larval development into the autoinfective filariform stage through mechanisms resembling molting signals, thereby accelerating the autoinfection cycle [18]. Our findings reinforce the importance of screening and preventive strategies in patients anticipated to receive corticosteroids in endemic areas. Prior studies evaluating prophylactic strategies, including randomized trials of thiabendazole and comparisons between targeted screening and empiric ivermectin, have not demonstrated a significant reduction in strongyloidiasis [19,20]. However, these studies were likely underpowered, and larger prospective studies are needed to define the optimal preventive approach. Based on our findings, clinicians in endemic areas should consider serologic or parasitologic screening for strongyloidiasis in all patients anticipated to initiate corticosteroids or other immunosuppressive therapies, with empiric ivermectin treatment given if screening is not feasible prior to treatment initiation.

Eosinophilia was present in only 40.3% of patients with uncomplicated infection, lower than the 50–70% reported in prior studies of chronic strongyloidiasis in otherwise healthy populations [21]. This difference likely reflects the hospitalized nature of our cohort, in which concurrent infections, systemic inflammation, and corticosteroid exposure may suppress eosinophil counts [17]. Importantly, eosinophilia was inversely associated with both severe disease and 30-day mortality. Severe cases had significantly lower absolute eosinophil counts, and moderate-to-severe eosinophilia was observed exclusively among uncomplicated cases. These findings are consistent with prior reports demonstrating that patients with eosinophilia have a more favorable prognosis [14]. This may reflect preserved host immune function, whereas corticosteroid exposure and severe systemic illness may suppress eosinophil production or promote eosinophil apoptosis [22].

Bacteremia was strongly associated with both severe strongyloidiasis and 30-day mortality in our cohort. These findings are consistent with prior studies demonstrating a higher frequency of bacteremia among non-survivors of severe strongyloidiasis [14]. This association reflects the pathophysiology of hyperinfection syndrome, in which larval migration across the intestinal mucosa facilitates translocation of enteric bacteria into the bloodstream [23]. In our cohort, gram-negative organisms predominated, particularly *K.pneumoniae* and *E.coli*; however, gram-positive organisms, including *Enterococcus* spp. and group D *Streptococcus*, as well as polymicrobial infections, were also observed, consistent with previous reports [23–25]. These findings highlight bacteremia as a key marker of systemic dissemination and underscore the importance of prompt antimicrobial therapy with coverage for both gram-negative and gram-positive enteric organisms, in addition to antiparasitic treatment.

The mortality rate among severe cases in our cohort (38.3%) was slightly lower than that reported in a recent meta-analysis (44.8%); however, this comparison should be interpreted cautiously, as that meta-analysis included cases defined under traditional hyperinfection and dissemination criteria, whereas our study applied a pragmatic clinical classification based on multi-organ involvement and specialist documentation, which may capture a somewhat different patient spectrum [14]. The meta-analysis predominantly included case reports and small case series, which are susceptible to publication bias and may overrepresent severe or fatal cases. Notably, a substantial proportion of patients in prior studies did not receive antiparasitic therapy, and mortality was extremely high among untreated individuals [14]. In contrast, all patients with severe disease in our cohort received antiparasitic treatment, which may have contributed to improved outcomes. Access to non-oral formulations of ivermectin may further improve outcomes in severe strongyloidiasis, particularly in patients with impaired gastrointestinal absorption [26].

Diagnosis of strongyloidiasis in endemic settings remains challenging, particularly among hospitalized patients with multiple comorbidities. Chronic infection is often asymptomatic or presents with nonspecific symptoms that may

overlap with common hospital-acquired conditions, such as nosocomial diarrhea, leading to underrecognition [21,27]. Severe strongyloidiasis is similarly difficult to identify, as clinical manifestations frequently mimic other causes of sepsis or respiratory failure, and characteristic features such as larva currens are uncommon [14]. Diagnostic sensitivity is further limited by the low yield of single stool examinations, particularly in patients with low parasite burden, whereas detection rates are higher in severe disease due to increased larval output. Although serologic testing has high reported sensitivity, its utility in endemic areas is limited by background seropositivity, cross-reactivity, and reduced performance in immunocompromised hosts [28–30]. Therefore, parasitologic confirmation remains essential, particularly in hospitalized patients with suspected symptomatic infection, where accurate diagnosis has important implications for clinical management.

Notably, after adjustment for physiologic instability and bacteremia, severe strongyloidiasis was not independently associated with 30-day mortality. One explanation is that the effect of severe disease may be mediated through downstream complications, such as bacteremia and hemodynamic instability, which are more direct determinants of mortality. Physiologic severity at presentation may therefore provide a more accurate assessment of mortality risk than parasitologic classification alone. In addition, early recognition and prompt initiation of antiparasitic therapy in our cohort may have mitigated the direct impact of severe strongyloidiasis on mortality. These findings suggest that outcomes in hospitalized patients are driven primarily by systemic complications rather than classification alone, highlighting the importance of early diagnosis and aggressive management of complications.

To our knowledge, this study represents one of the largest cohorts of parasitologically confirmed strongyloidiasis in Southeast Asia and provides clinically relevant data on predictors of severe disease and mortality in hospitalized patients. However, Several limitations should be considered. First, selection bias may be present because our study included only hospitalized patients with parasitologically confirmed infection. Patients with mild or undiagnosed infection were likely underrepresented, potentially resulting in overrepresentation of clinically apparent or severe cases. In addition, stool examination was performed based on clinical indication and has limited sensitivity, which may have further contributed to underdiagnosis of uncomplicated infection. Second, the severe case definition required infectious diseases specialist documentation, which may introduce ascertainment bias if consultation patterns differed between groups; however, this was applied alongside objective microbiologic evidence and within a setting where specialist consultation is routinely performed for severe infections. Third, the retrospective design limits causal inference and introduces risk of incomplete data capture. For example, detection of *Strongyloides stercoralis* in patients evaluated for diarrhea does not necessarily establish causality, as diarrhea in hospitalized patients is often multifactorial. Fourth, this was a single-center study, which may limit generalizability. Fifth, detailed information on corticosteroid dose, duration, and indication was not systematically captured in the electronic database, precluding dose-response analysis; future studies should prospectively collect these data to better characterize the threshold exposure associated with severe disease. Finally, residual confounding cannot be excluded despite multivariable adjustment.

In conclusion, recent corticosteroid exposure was the strongest independent predictor of severe strongyloidiasis, while eosinophilia was associated with lower risk of severe disease and mortality. Bacteremia and physiologic instability, rather than disease classification alone, were the primary determinants of short-term mortality. These findings emphasize the importance of early recognition and prompt treatment, particularly in immunosuppressed patients, to improve clinical outcomes in endemic settings.

## Supporting information

**S1 Table. Detailed Clinical Manifestations and Complications Among Patients with Severe Strongyloidiasis.**
Detailed demographic data, underlying conditions, clinical manifestations, and organ involvement among patients classified as having severe strongyloidiasis.
(DOCX)

**S2 Table. Microbiologic Characteristics of Bacteremia in Severe Strongyloidiasis.** Details of bloodstream infections, including causative organisms, frequency of gram-negative and gram-positive pathogens, and occurrence of polymicrobial infections.
(DOCX)

**S3 Table. Distribution of Larval Detection by Specimen Type in Severe Strongyloidiasis.** Types and frequencies of clinical specimens yielding *Strongyloides stercoralis* larvae, including stool, respiratory samples, gastrointestinal specimens, and sterile body fluids.
(DOCX)

**S4 Table. Radiographic Findings in Severe Strongyloidiasis with Abnormal Chest Imaging.** Summary of chest radiographs findings among patients with severe disease.
(DOCX)

**S1 Dataset. Anonymized patient-level data for all hospitalized patients with laboratory-confirmed *Strongyloides stercoralis* infection used in this analysis.**
(XLSX)

## Author contributions

**Conceptualization:** Sorawit Chittrakarn.

**Data curation:** Sorawit Chittrakarn, Nonthanat Tongsengkee, Nattapat Sangkakul, Siripen Kanchanasuwan.

**Formal analysis:** Sorawit Chittrakarn, Siripen Kanchanasuwan.

**Methodology:** Sorawit Chittrakarn, Nonthanat Tongsengkee, Nattapat Sangkakul, Siripen Kanchanasuwan.

**Project administration:** Sorawit Chittrakarn.

**Software:** Sorawit Chittrakarn.

**Supervision:** Sorawit Chittrakarn, Siripen Kanchanasuwan.

**Validation:** Sorawit Chittrakarn, Nonthanat Tongsengkee.

**Visualization:** Sorawit Chittrakarn, Siripen Kanchanasuwan.

**Writing – original draft:** Sorawit Chittrakarn, Nonthanat Tongsengkee.

**Writing – review & editing:** Sorawit Chittrakarn, Nonthanat Tongsengkee, Nattapat Sangkakul, Siripen Kanchanasuwan.

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
