## [Decision Letter · Decision Letter 0]

3 Apr 2026

Response to Reviewers'. This file does not need to include responses to any formatting updates and technical items listed in the 'Journal Requirements' section below.'. This file does not need to include responses to any formatting updates and technical items listed in the 'Journal Requirements' section below.* A marked-up copy of your manuscript that highlights changes made to the original version. You should upload this as a separate file labeled 'Revised Manuscript with Track Changes'.'.* An unmarked version of your revised paper without tracked changes. You should upload this as a separate file labeled 'Manuscript'.'.If you would like to make changes to your financial disclosure, competing interests statement, or data availability statement, please make these updates within the submission form at the time of resubmission. Guidelines for resubmitting your figure files are available below the reviewer comments at the end of this letter.We look forward to receiving your revised manuscript.Kind regards,David J. Diemert, M.D.Academic EditorPLOS Neglected Tropical DiseasesJong-Yil ChaiSection EditorPLOS Neglected Tropical Diseases

Shaden Kamhawi

co-Editor-in-Chief

Paul Brindley

co-Editor-in-Chief

**Journal Requirements:**

At this stage, the following Authors/Authors require contributions: Sorawit Chittrakarn, Nonthanat Tongsengkee, Nattapat Sangkakul, and Siripen Kanchanasuwan. Please ensure that the full contributions of each author are acknowledged in the "Add/Edit/Remove Authors" section of our submission form.

2) We notice that your supplementary Tables are included in the manuscript file. Please remove them and upload them with the file type 'Supporting Information'. Please ensure that each Supporting Information file has a legend listed in the manuscript after the references list.

**Reviewers' comments:**Reviewer's Responses to Questions

**Key Review Criteria Required for Acceptance?**

**Methods:**

-Are the objectives of the study clearly articulated with a clear testable hypothesis stated?

-Is the study design appropriate to address the stated objectives?

-Is the population clearly described and appropriate for the hypothesis being tested?

-Is the sample size sufficient to ensure adequate power to address the hypothesis being tested?

-Were correct statistical analysis used to support conclusions?

-Are there concerns about ethical or regulatory requirements being met?

Reviewer #1: The methods is appropriate for research question.

Reviewer #2: 1. The definition of severe strongyloidiasis requiring infectious disease specialist documentation introduces potential ascertainment bias.

2. No sample size calculation or power analysis is provided.

3. The handling of missing data is not described.

4. Line 160-161 contains an incomplete/redundant sentence ("clinically relevant a priori were included in the multivariable model").

5. Model discrimination (AUC) is mentioned but not reported in results.

- Recommendation: Major revision—address missing data handling, report AUC values, and correct the incomplete sentence.

**Results:**

-Does the analysis presented match the analysis plan?

-Are the results clearly and completely presented?

-Are the figures (Tables, Images) of sufficient quality for clarity?

Reviewer #1: Corticosteroid exposure - Please provide detail

- Dose and duration of steroid use.

- Is the indication of steroid use associate with Strongyloidiasis?

- Is steroid prescription with other immunosuppressor?

Reviewer #2: 1. The absence of chronic kidney disease in severe cases (0% vs 10.9%) is unexplained and counterintuitive—this warrants discussion.

2. Model fit statistics and AUC values are not reported despite being mentioned in methods.

3. The indication for stool examination in uncomplicated cases is informative but somewhat tangential to the main analysis.

4. Lack of specific corticosteroid dose exposure

- Recommendation: Minor revision—report model discrimination metrics and address the CKD finding. Also add the dose exposure of corticosteroids if data is available

**Conclusions:**

-Are the conclusions supported by the data presented?

-Are the limitations of analysis clearly described?

-Do the authors discuss how these data can be helpful to advance our understanding of the topic under study?

-Is public health relevance addressed?

Reviewer #1: The conclusion is appropriate.

Reviewer #2: 1. The discussion of HTLV-1 absence, while relevant, is somewhat lengthy given it was not a primary study objective.

2. The clinical implications for screening strategies could be more concrete—what specific recommendations emerge from these findings?

3. The comparison of mortality rates to prior meta-analyses should acknowledge differences in case definitions.

- Recommendation: Minor revision—strengthen clinical recommendations and streamline HTLV-1 discussion.

**Editorial and Data Presentation Modifications?**

Reviewer #1: Minor Revision

Reviewer #2: Minor revisions

**Summary and General Comments**

Reviewer #1: The overall research question, methods, results, and conclusions are well presented. I have only minor comments and questions in the Methods section. Additionally, the English language may require minor revision.

Reviewer #2: This is a well-conducted retrospective cohort study addressing an important clinical question in a neglected tropical disease. The large sample size, systematic data collection, and multivariable analyses are strengths. The key findings regarding corticosteroid exposure, eosinophilia, and the mediating role of bacteremia in mortality are clinically relevant and advance understanding of severe strongyloidiasis.

Tentative Decision: Accept with Minor Revisions

The manuscript requires correction of the incomplete sentence in methods, reporting of model discrimination metrics, and clarification of the unexpected CKD finding. These are addressable issues that do not undermine the study's validity or conclusions.

PLOS authors have the option to publish the peer review history of their article (what does this mean?). If published, this will include your full peer review and any attached files.). If published, this will include your full peer review and any attached files.). If published, this will include your full peer review and any attached files.). If published, this will include your full peer review and any attached files.

...

Reviewer #1: **Yes:** Pornpan KoomanachaiPornpan KoomanachaiPornpan KoomanachaiPornpan Koomanachai

Reviewer #2: **Yes:** Andres Henao-MartinezAndres Henao-MartinezAndres Henao-MartinezAndres Henao-Martinez

**Figure resubmission:**While revising your submission, we strongly recommend that you use PLOS’s NAAS tool (https://ngplosjournals.pagemajik.ai/artanalysis) to test your figure files. NAAS can convert your figure files to the TIFF file type and meet basic requirements (such as print size, resolution), or provide you with a report on issues that do not meet our requirements and that NAAS cannot fix.

**Reproducibility:**To enhance the reproducibility of your results, we recommend that authors of applicable studies deposit laboratory protocols in protocols.io, where a protocol can be assigned its own identifier (DOI) such that it can be cited independently in the future. Additionally, PLOS ONE offers an option to publish peer-reviewed clinical study protocols. Read more information on sharing protocols at https://plos.org/protocols?utm_medium=editorial-email&utm_source=authorletters&utm_campaign=protocolsTo enhance the reproducibility of your results, we recommend that authors of applicable studies deposit laboratory protocols in protocols.io, where a protocol can be assigned its own identifier (DOI) such that it can be cited independently in the future. Additionally, PLOS ONE offers an option to publish peer-reviewed clinical study protocols. Read more information on sharing protocols at https://plos.org/protocols?utm_medium=editorial-email&utm_source=authorletters&utm_campaign=protocols

---

## [Decision Letter · Decision Letter 1]

12 Apr 2026

Dear Mr. Chittrakarn,

We are pleased to inform you that your manuscript 'Predictors of Severe Strongyloidiasis and Mortality in Hospitalized Patients from Southern Thailand' has been provisionally accepted for publication in PLOS Neglected Tropical Diseases.

Best regards,

David J. Diemert, M.D.

Academic Editor

jong-Yil Chai

Section Editor

Shaden Kamhawi

co-Editor-in-Chief

Paul Brindley

co-Editor-in-Chief

Reviewer's Responses to Questions

**Key Review Criteria Required for Acceptance?**

**Methods**

-Are the objectives of the study clearly articulated with a clear testable hypothesis stated?

-Is the study design appropriate to address the stated objectives?

-Is the population clearly described and appropriate for the hypothesis being tested?

-Is the sample size sufficient to ensure adequate power to address the hypothesis being tested?

-Were correct statistical analysis used to support conclusions?

-Are there concerns about ethical or regulatory requirements being met?

Reviewer #1: Method is appropriate

Reviewer #2: (No Response)

**Results**

-Does the analysis presented match the analysis plan?

-Are the results clearly and completely presented?

-Are the figures (Tables, Images) of sufficient quality for clarity?

Reviewer #1: The authors demonstrated relevant results and tried to revised as comments.

Reviewer #2: (No Response)

**Conclusions**

-Are the conclusions supported by the data presented?

-Are the limitations of analysis clearly described?

-Do the authors discuss how these data can be helpful to advance our understanding of the topic under study?

-Is public health relevance addressed?

Reviewer #1: Appropriate

Reviewer #2: (No Response)

**Editorial and Data Presentation Modifications?**

Reviewer #1: Accept

Reviewer #2: (No Response)

**Summary and General Comments**

Reviewer #1: The research is benefit for publication and Knowledgeable.

Reviewer #2: (No Response)

PLOS authors have the option to publish the peer review history of their article (what does this mean?). If published, this will include your full peer review and any attached files.). If published, this will include your full peer review and any attached files.). If published, this will include your full peer review and any attached files.). If published, this will include your full peer review and any attached files.

...

Reviewer #1: **Yes:** Pornpan KoomanachaiPornpan KoomanachaiPornpan KoomanachaiPornpan Koomanachai

Reviewer #2: **Yes:** Andrés F. Henao-MartínezAndrés F. Henao-MartínezAndrés F. Henao-MartínezAndrés F. Henao-Martínez

---

## [Editor Report · Acceptance letter]

Dear Mr. Chittrakarn,

We are delighted to inform you that your manuscript, "Predictors of Severe Strongyloidiasis and Mortality in Hospitalized Patients from Southern Thailand," has been formally accepted for publication in PLOS Neglected Tropical Diseases.

Best regards,

Shaden Kamhawi

co-Editor-in-Chief

Paul Brindley

co-Editor-in-Chief
